# Morphometry of Boar Spermatozoa in Semen Stored at 17 °C—The Influence of the Staining Technique

**DOI:** 10.3390/ani12151888

**Published:** 2022-07-25

**Authors:** Dorota Szablicka, Anna Wysokińska, Angelika Pawlak, Klaudia Roman

**Affiliations:** Faculty of Agrobioengineering and Animal Husbandry, Siedlce University of Natural Sciences and Humanities, 08110 Siedlce, Poland; dorotaszablicka@o2.pl (D.S.); pawlak045@gmail.com (A.P.); klaudia8c@gmail.com (K.R.)

**Keywords:** boar, semen storage time, sperm morphometry

## Abstract

**Simple Summary:**

To obtain satisfactory results in artificial insemination, it is necessary to use high-quality ejaculates for the production of insemination doses and then maintain the biological value of the sperm during storage. Boar spermatozoa, owing to the specific structure of the cell membrane, are particularly sensitive to temperature changes. For this reason, cryopreservation cannot be used in artificial insemination practice, and there may be limitations to successful storage of semen in a liquid state. The practice of using boar semen for artificial insemination does not include analyses of the effect of storage time of boar semen on sperm dimensions. Therefore, the aim of the study was to analyse the morphometry of sperm during storage of liquid boar semen. An attempt was also made to evaluate the suitability of three staining methods for assessment of boar sperm morphometry. The morphometric dimensions of boar sperm were shown to change during storage of liquid semen. These changes affected the sperm head more than the tail and were due to the staining method used. The analyses are very important as they provide more information about the morphometric dimensions of the sperm during preservation of boar semen. The applied sperm staining techniques allows for a more accurate assessment of male reproductive cells.

**Abstract:**

The aim of the study was to assess the morphometry of sperm during storage of liquid boar semen at 17 °C. An attempt was also made to evaluate the suitability of three staining methods for assessment of boar sperm morphometry. The study was carried out on 20 Landrace boars. Semen was collected from the boars every 5 days by the manual method. Four ejaculates from each boar were analysed (80 ejaculates in total). Analyses were performed five times: at 1 h, 24 h, 48 h, 96 h, and 168 h after semen collection. Blisters with insemination doses were opened immediately before the analyses. From each insemination dose, smears were prepared for morphometric evaluation of sperm, which were stained by three methods (eosin-nigrosin—EN, eosin-gentian—EG, and SpermBlue—SB). Morphometric measurements of 15 randomly selected sperm with normal morphology were performed on each slide. The morphometric measurements included the following parameters: sperm head length, width, area, and perimeter; tail length; and total sperm length. The results of the morphometric measurements were used to calculate the head shape index. The morphometric dimensions of the sperm were shown to change during storage of semen at 17 °C. The extent of these changes, however, depended on the staining method used, as the three methods result in different morphometric dimensions of sperm, in the case of both the head and the tail. In the slides stained by the eosin-nigrosin method, the dimensions of the head and tail were smaller at every time of storage than in the slides stained by the SpermBlue and eosin-gentian methods.

## 1. Introduction

Artificial insemination of pigs is mainly carried out using semen in liquid form, usually stored at 15–17 °C [1]. Storage of diluted boar semen at 15–17 °C decreases the metabolism of sperm, which is essential for preserving semen quality [2,3,4]. In order to obtain satisfactory results in artificial insemination, it is necessary to use high-quality ejaculates for the production of insemination doses and then maintain the biological value of the sperm during storage. Boar spermatozoa, owing to the specific structure of the cell membrane, are particularly sensitive to temperature changes. For this reason, cryopreservation is rarely used in artificial insemination practice, and there may be limitations to successful storage of semen in liquid state. Thanks to its high content of polyunsaturated fatty acids [5] and low cholesterol level relative to phospholipids [6,7], the cell membrane of boar sperm is more susceptible to damage than that of sperm of other mammalian species. This can cause significant variation in the predispositions of individuals for use for artificial insemination. Most diluted insemination doses are used for artificial insemination on the day they are collected, but some are used several days later [7]. Dilution of boar semen is believed to reduce proteins and natural antioxidants together with other components of seminal plasma, which are essential for the proper functioning and integrity of sperm membranes [8]. During storage of boar semen, functional and structural changes take place in sperm, which can be affected by the dilution conditions and storage time [9]. For this reason, it is important to monitor boar semen during storage to the extent possible. Routine ejaculate evaluation is often insufficient to predict whether semen parameters will change during preservation. The morphometric dimensions of sperm have been shown to be associated with the fertility of the boar [10,11]. Development of criteria for morphometric assessment of individual sperm structures can also facilitate quality control of stored semen. In the case of boars, even sperm staining itself proves to be problematic [12]. This results in unsatisfactory visualization of specific sperm structures, which makes it difficult to objectively and accurately assess their size and shape. In addition, in morphometric evaluation, it is very important to use a suitable staining technique with minimal impact, or none at all, on sperm structure. Methods recommended by the WHO for staining of human semen include the Papanicolau and Diff–Quik methods, and previously the eosin-nigrosin method. In veterinary medicine, there are no strict guidelines regarding the use of staining techniques, although there are certain preferences in research on various animal species. Staining methods used for analysis of the morphological characteristics of sperm of various species of animals include eosin-nigrosin [13], eosin-gentian [12], SpermBlue [14], Giemsa stain [15], Diff-Quik [16,17], and carbolfluchsin-eosin [18]. The staining method that quite clearly marks the sperm head and tail is the eosin-gentian method, which we used in our previous studies on boar semen [12], stallions [19], and dogs [17]. In turn, the SpermBlue method is a relatively simple method, suitable for field conditions, that successfully stains human and animal sperm of various species in different shades of blue intensities [14]. It is also often used to assess sperm morphometry using the SCA (*sperm class analyzer*). The eosin-nigrosin system is an easy-to-perform method. It does not require fixation in any buffer or alcohol, and is used to assess live and dead sperm [20]. It also allows the elimination of dead sperm, the heads of which are pink. For boar semen, unfortunately, no suitable staining method has been indicated so far, and no clear criteria standardizing morphometric measurements in this species have been developed. Moreover, the effect of storage time of liquid boar semen on sperm dimensions has not been analysed. However, it has been demonstrated that processing of the ejaculate after collection can cause changes in the cellular structures of sperm and, therefore, affect their capacity to fertilize the ovum [21,22,23].

The aim of the study was to assess the morphometry of sperm during storage of liquid boar semen. An attempt was also made to evaluate the suitability of three staining methods for the assessment of boar sperm morphometry.

## 2. Materials and Methods

The study was carried out in strict compliance with the recommendations in Directive 63/2010/EU and the Journal of Laws of the Republic of Poland of 2015 on the pro-tection of animals used for scientific or educational purposes. The study was approved by the Polish Laboratory Animal Science Association (Number 3401/2015).

### 2.1. Animal and Semen Collection

The study was conducted on 20 Landrace boars routinely used at a sow insemination station with proven field fertility. Boars at the age of 18–24 months were selected for the study. All boars were healthy and kept in appropriate welfare conditions. Semen was collected from the boars by the manual method every 5 days. Four ejaculates per boar were used for the analysis (80 in total). Ejaculate volume and sperm motility and concentration were determined for each ejaculate. Ejaculate volumes were determined after filtrating the gelatinous fraction off, using electronic scales to measure ejaculate weight. Sperm motility was evaluated using a Nikon Eclipse 50i light microscope equipped with a heated stage. A sample of 5 μL of sperm suspension was placed on a pre-warmed slide and sealed with a coverslip at 37 °C. Using 200× magnification, the percentage of normally motile spermatozoa was determined based on the number of sperm present in the field of vision of the microscope. The concentration of spermatozoa was determined by the colorimetric method using an AccuRead photometer (IMV Technologies, L’Aigle, France). Ejaculates in which sperm motility was at least 70% were diluted with Biosolwens Plus extender (Biochefa, Sosnowiec, Poland) so that each insemination dose contained 2.9 × 10^9^ sperm. Diluted ejaculates were packed in 90 ml plastic blisters and stored in an insulated box at 17 °C.

### 2.2. Analyses and Slide Preparation

The analyses were performed five times: at 1 h, 24 h, 48 h, 96 h, and 168 h after semen collection. The first evaluation was performed 1 h after diluting the ejaculate and cooling the sample to 17 °C. Blisters with insemination doses were opened immediately before the analyses. From each insemination dose, slides were prepared for morphometric evaluation of sperm.

Microscope slides were prepared by three staining methods: eosin-nigrosin (EN), eosin-gentian (EG), and SpermBlue (SB).

Eosin-nigrosine staining was performed according to the method described by Wysokińska et al. [17]. The procedure for staining eosin-gentian is presented in Wysokińska and Kondracki [24]. SpermBlue staining was based on the method presented by van der Horst and Maree [14]. A thin semen smear was spread on a microscope slide heated to 37 °C. After drying, the slide was stained for about 15 min with SpermBlue stain (Microptic SL, Barcelona, Spain). Then, it was rinsed in distilled water and air-dried.

### 2.3. Morphometric Measurements

Microscope examination of the slides was carried out using a Nikon E-50i microscope (Tokyo, Japan), aDSFi3 digital camera, and NIS-elements D5 image analysis software. The measurements were made using a Wacom Intuos 6.3.17 drawing tablet with a highly sensitive digital pen (Krefeld, Germany). Images of sperm were analysed on a computer monitor. On each slide, morphometric measurements were taken of 15 randomly selected sperm with normal morphology (showing no head defects, cytoplasmic droplets, or coiled tails). In total, measurements of 18,000 sperm were taken. The morphometric measurements included the following parameters: sperm head length, width, area, and perimeter; tail length; and total sperm length [24].

Based on the results of the morphometric measurements, the shape indices of the sperm head were calculated: ellipticity (length/width), elongation (length − width)/(length + width), rugosity (4π × area/perimeter^2^), and regularity (π × length × width /4 × area).

### 2.4. Statistical Analysis

Statistical analysis of the results was performed using Statistica v.13.1 software (StatSoft, Tulsa, OK, USA) using analysis of variance. The results are presented as mean ± standard deviation (SD). Statistical analysis of the material was performed according to the following mathematical model: Y*_ijk_* = µ + a*_i_* + b*_j_* + ab*_ij_* + e*_ijk_*, where Y*_ijk_*—value of trait, µ– population mean; a*_i_*—factor 1 (semen storage time); b*_j_*—factor 2 (staining method); ab*_ij_*—the effect of interaction; and e*_ijk_*—error. The significance between the groups was determined by Tukey’s test (*p* ≤ 0.05).

## 3. Results

Table 1 presents the morphometric dimensions of sperm stained by the eosin-nigrosin method in semen stored at 17 °C. The sperm dimensions were shown to have changed during storage of diluted semen. The morphometric dimensions of both the head and tail of the sperm were shown to have decreased within the first 24 h of storage. In subsequent hours of semen storage, there was a gradual increase in the head dimensions, i.e., its length, width, area, and perimeter, as well as in the length of the tail and of the entire spermatozoon. The greatest increase in the dimensions of the sperm head were noted between 24 and 48 h of semen storage, with significant differences (*p* ≤ 0.05) shown in the case of the length, width, and perimeter of the sperm head. The length of the tail and of the entire spermatozoon increased from 24 h of semen storage. At 168 h of semen storage, the total sperm length was 51.85 µm and it was 3.82 µm greater than at 24 h of storage (*p* ≤ 0.05). The lowest values for the ellipticity and elongation of the sperm head were shown at 168 h of semen storage.

The average values for the morphometric dimensions of sperm stained by the eosin-gentian method in semen stored at 17 °C are presented in Table 2. The data indicate that the head dimensions of sperm stained by the eosin-gentian method increase up to 96 h of semen storage and then decrease. Significant differences were noted between the first hour and 48 and 96 h of storage (*p* ≤ 0.05). The tail dimensions and the total sperm length did not change with the semen storage time. No differences were shown for the ellipticity or elongation of the sperm heads, but the differences in the other head shape indices between 1 h of semen storage and the other times were confirmed statistically.

Table 3 presents the results for the morphometry of sperm stained by the SpermBlue method in semen stored at 17 °C. The data indicate that the sperm head dimensions showed little variation depending on semen storage time. Sperm length in the first 24 h of storage was 0.37 µm lower than at 96 h of storage (*p* ≤ 0.05). The heads were widest at 48 h of semen storage (on average 5.02 µm). The area and perimeter of the sperm heads were smaller after 24 h of semen storage than at 1, 48, 96, and 168 h (*p* ≤ 0.5). The semen storage time was not shown to affect the tail dimensions or the head shape indices.

Figure 1 presents the morphometric dimensions of sperm heads in each of the three staining methods (EN, EG, and SB) at different times of semen storage. Analysis of the results presented in the graphs reveals distinct differences in the morphometry of sperm stained by the three staining methods at different times of semen storage. In the case of sperm stained by each of the three methods, the length of the sperm heads increased up to 96 h of storage and then decreased (Figure 1A). At each time of semen storage, significant differences were noted for the head length of sperm stained by the EN, EG, and SB methods (*p* ≤ 0.05). The lowest head length was noted for sperm stained by the eosin-nigrosin method at every semen storage time. Sperm stained by the eosin-gentian method had longer heads (*p* ≤ 0.05) than sperm stained by the SpermBlue and eosin-nigrosin methods.

The values for the head width of sperm stained by the three staining techniques are presented in Figure 1B. Sperm stained by the eosin-nigrosin method had the narrowest (*p* ≤ 0.05) heads up to 96 h. Sperm stained by the eosin-gentian method had the widest heads at 24, 48, and 96 h of storage (*p* ≤ 0.05). The width of the sperm heads was the same in all techniques only at 168 h of semen storage.

Figure 1C presents the average surface area of sperm heads depending on the staining method at each semen storage time. The data clearly indicate that sperm stained by the eosin-nigrosin method have the smallest head area (*p* ≤ 0.05). The area of sperm heads was the largest in semen stained by the eosin-gentian method at every time of preservation (*p* ≤ 0.05).

The mean values for the perimeter of sperm heads stained by the three staining methods at each time of storage are presented in Figure 1D. The data show that sperm stained by the eosin-nigrosin method had the smallest (*p* ≤ 0.05) head perimeter than in the case of the other methods. For sperm stained by the eosin-gentian and SpermBlue techniques, the head perimeter showed a similar tendency to increase with storage time.

Analysis of the tail length of sperm and the total sperm length reveals that, in semen stained by the eosin-nigrosin method, the sperm had smaller dimensions than in the case of the eosin-gentian and SpermBlue (*p* ≤ 0.05) (Figure 2A,B). In addition, in semen stained by the eosin-nigrosin method, these measurements showed clear upward trends with the passage of semen storage time. Sperm stained by the eosin-gentian and SpermBlue methods had similar tail length and sperm length after 24, 48, and 168 h of storage of diluted ejaculates.

## 4. Discussion

The results of the study indicate that the morphometric dimensions of sperm change during storage of boar semen at 17 °C. The extent of these changes, however, varies depending on the staining method used. The staining methods used in this study (EN, EG, and SB) affect the morphometric dimensions of both the head and tail of sperm. The slides stained by the eosin-nigrosin method showed smaller head and tail dimensions than in the case of the SpermBlue and eosin-gentian methods at every hour of semen storage. Similar observations were reported by Kondracki et al. [12]. The present study also showed that the sperm head dimensions obtained on the basis of measurements performed on slides stained by the eosin-gentian method were larger than in the case of the other two staining methods. Only in the measurements made at 168 h of semen storage were the results for head width similar to those obtained in the other staining methods. According to some authors, the morphometric dimensions of sperm may be influenced by the duration of staining [25]. The work of these authors showed that the intensity and contrast of the images can be improved by extending the staining time. This resulted in larger morphometric dimensions of sperm.

A major problem in veterinary medicine is the lack of standardization in the choice of staining technique for the evaluation of sperm morphology and morphometry. Many researchers have compared the effects of various staining methods on sperm morphology and morphometry. Some have shown differences in the dimensions of the head and midpiece using different staining methods [25,26]. For example, the recommended staining method for human semen is Papanicolaou staining. However, this method is time-consuming and does not effectively stain animal sperm [27,28]. Another method that has been used for the assessment of the morphology and morphometry of human and animal sperm is Diff-Quik. However, studies using human semen have shown that this method causes the sperm head to swell [29]. According to Chacon [30], carbol-fuchsin staining can be a method supplementing the evaluation of sperm morphology in unstained smears. Recently, the SpermBlue method, recommended for staining human and animal semen, is increasingly being used [14]. Research using boars has shown that, in sperm stained by the SpermBlue method, the dimensions of the sperm head were the most similar to those of unstained sperm [26]. In that study, other staining methods such as Papanicolaou caused the sperm heads to shrink, whereas AgNO_3_ staining caused them to swell. In the present study, the sperm head dimensions were similar in the case of staining with SpermBlue and eosin-gentian. Thus, our results and literature data seem to indicate that these methods can be used in the diagnosis of boar semen. Nevertheless, in considering the effect of the staining method on sperm morphology, the species of animal should be considered, because a method that stains the sperm of one species well and without negative effects may not be suitable for another species.

Apart from the effect of the staining technique, sperm morphometry can be influenced by other factors, such as the slide preparation procedure, the size of the microscope lens, or even the person performing the test [19,31,32,33], as well as the pH of the stains, osmotic concentration, and staining duration [34]. In a study on fresh stallion semen, Gacen et al. [34] observed no differences in the percentage of sperm with abnormal morphology using eosin-nigrosin staining and Trumorph^®^.

In the present study, we used three staining methods. The eosin-nigrosin method is used for the evaluation of semen in both veterinary and human medicine [20,35]. A study by Łukaszewicz et al. [33], using four different methods based on the use of eosin and nigrosin, showed that eosin-nigrosin staining is a simple, inexpensive, and non-invasive method for the evaluation of the quality of poultry semen, and thus for the assessment of the fertilizing capacity of sperm. The study also showed that the low osmotic pressure of the stain relative to semen causes the sperm to swell. Another study, using bulls, showed that the eosin-nigrosin method is suitable for the evaluation of the sperm head and acrosomal cap [36]. The eosin-gentian staining method is used mainly in veterinary medicine for the evaluation of sperm morphology and morphometry [12,19,27]. This method provides a clear image of the cells, but can cause swelling of the sperm heads [37]. A staining technique that has recently become very popular is the SpermBlue method, which has already been used for the evaluation of animal sperm [14,17,38,39]. The method of staining should not affect the cellular structures of sperm, but it should stain its individual elements in a way that allows their accurate assessment [40]. The choice of a suitable staining technique is particularly important during semen diagnosis performed during the preservation of boar semen, as boar sperm are highly sensitive to storage conditions. During storage of boar semen, reorganization of lipids in the cell membrane of sperm may take place, leading to destabilization of the membrane. These changes can affect the functions of individual sperm structures [41,42,43]. It was indicated that, during the storage of liquid boar semen, functional changes occur in sperm cells, reminiscent of the natural aging processes of sperm [44]. These changes may include various sperm structures, including the plasma membrane [20] or mitochondria [45]. The mechanism of sperm aging during storage is related to, inter alia, lipid peroxidation and changes in the fluidity of the sperm membrane initiating premature capacitation [46]. Therefore, it is possible that these changes can appear in the dimensions of the sperm cells.

In the present study, the staining method was shown to affect morphometric evaluations of sperm carried out at various semen storage times. The most similar results were obtained in the morphometric evaluation of sperm using the eosin-gentian and SpermBlue methods. This may be because of their similar slide preparation procedures, as in both cases, the smears are fixed and then stained.

Many studies have analysed the morphometric features of sperm of various animal species [47,48,49] in an attempt to find connections between sperm dimensions and male fertility [50,51]. Sperm with smaller heads have been shown to be more effective at fertilizing the ovum [32], have acrosome-intact [52], and to result in larger litters [53]. The main element of the sperm head is the cell nucleus, which contains chromatin. Chromatin packaging in the sperm cell nucleus involves protamination, i.e., replacement of histones by protamines [54,55]. Replacing the histones with protamines reduces the chromatin area in the sperm nucleus. Therefore, it can be concluded that abnormalities in the structure and shape of the spermatozoa may be related to a disturbed structure of chromatin [56]. Urbano et al. [57] analysed four subpopulations of sperm and found that small sperm heads had a higher rate of DFI. In addition, sperm with round heads are slower than sperm with elongated heads [58]. Given these observations, it seems that the dimensions of the sperm heads are important in evaluating male gametes in terms of their fertilization capacity. In our study, at 168 h of storage of liquid semen, the width of the sperm head was similar in the case of all staining methods used.

Most studies on sperm morphometry have dealt mainly with the sperm head, because this structure is measured using CASA systems. Few studies have evaluated the sperm tail. This part of the spermatozoon is an important structure because it is responsible for its motility. Sperm with a longer midpiece and tail have been shown to be stronger in the tail, allowing them to reach the ovum quicker [59]. In the present study, the length of the tail and of the whole sperm did not change during semen storage. This was confirmed by the results obtained in slides stained by the SpermBlue, eosin-gentian, and eosin-nigrosin methods. The sperm midpiece is an important structure because it contains a densely packed spiral of mitochondria [60], which can supply the large amount of energy needed for sperm motility through oxidative phosphorylation [61,62]. The fertilization mode is of key importance in the evolution of sperm length in animals. It is influenced by the environment in which the fertilization process takes place [63]. Moreover, sperm length in insects has been shown to evolve faster in fertile sperm [64]. Sperm length should be determined as the sum of the tail length and head length. In our study, sperm length did not change during the storage of liquid boar semen.

## 5. Conclusions

To conclude, the morphometric dimensions of boar sperm cells undergo changes during the storage of liquid boar semen. These changes affect the sperm head more than the tail. It can thus be assumed that, during the storage of liquid boar semen, changes occur in the sperm head, which may affect their fertilizing ability. The EG, EN, and SB staining methods used show changes in the morphometric dimensions of boar sperm during semen conservation. The most similar results are obtained when using EG and SB staining. There is a need for further analyses on the morphometry of sperm during preservation, including boars of different breeds, with simultaneous fertility analysis.

## Figures and Tables

**Figure 1 animals-12-01888-f001:**
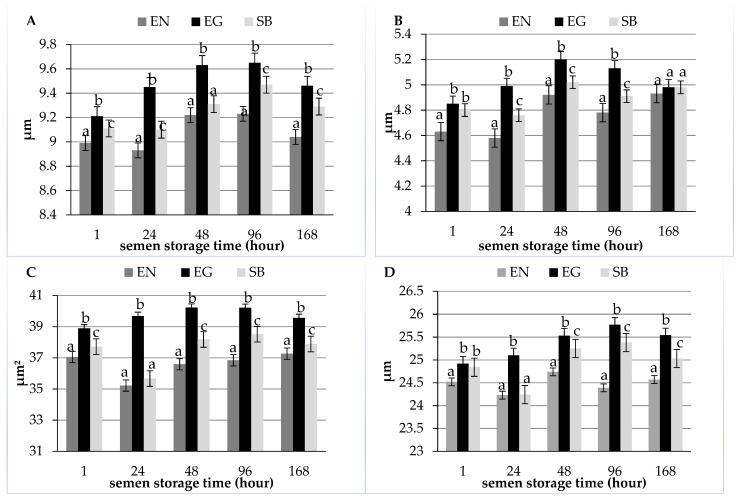
Morphometric dimensions of the heads of sperm stained by the eosin-nigrosin (EN), eosin-gentian (EG), and SpermBlue (SB) methods depending on semen storage time ((**A**)—length, (**B**)—width, (**C**)—area, (**D**)—perimeter). Bars with different letters mean statistically significant values (*p* ≤ 0.05).

**Figure 2 animals-12-01888-f002:**
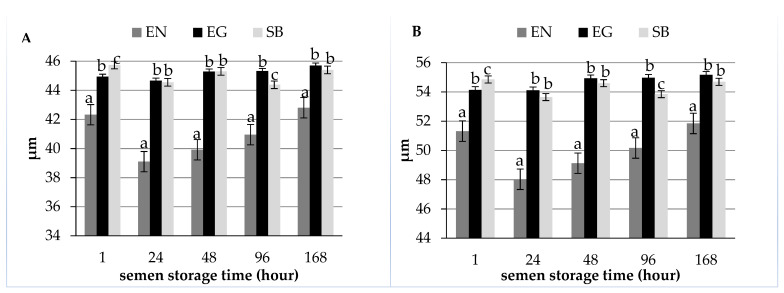
Length of sperm stained by the eosin-nigrosin (EN), eosin-gentian (EG), and SpermBlue (SB) methods depending on semen storage time ((**A**)—tail length, (**B**)—total sperm length). Bars with different letters mean statistically significant values (*p* ≤ 0.05).

**Table 1 animals-12-01888-t001:** Morphometric dimensions of sperm stained by the eosin-nigrosin method in semen stored at 17 °C (mean ± SD).

Item	Semen Storage Time (Hour)				
	1	24	48	96	168
Number of analyzed cells	1200	1200	1200	1200	1200
Head					
Length (µm)	8.99 ± 0.88 ^a^	8.93 ± 0.87 ^a^	9.22 ± 0.85 ^b^	9.23 ± 0.76 ^b^	9.04 ± 0.76 ^ab^
Width (µm)	4.63 ± 0.57 ^a^	4.58 ± 0.61 ^a^	4.92 ± 0.66 ^b^	4.78 ± 0.65 ^b^	4.93 ± 0.58 ^b^
Area (µm^2^)	37.05 ± 4.96 ^a^	35.22 ± 3.86 ^b^	36.59 ± 4.44 ^ab^	36.84 ± 4.23 ^ab^	37.27 ± 4.12 ^a^
Perimeter (µm)	24.52 ± 1.81 ^ab^	24.23 ± 1.61 ^a^	24.74 ± 1.76 ^b^	24.39 ± 1.71 ^ab^	24.57 ± 1.58 ^ab^
Tail					
Length (µm)	42.32 ± 5.45 ^a^	39.10 ± 6.42 ^b^	39.92 ± 6.29 ^bc^	40.95 ± 7.06 ^c^	42.80 ± 6.34 ^a^
Sperm total length (µm)	51.32 ± 5.67 ^ac^	48.03 ± 6.53 ^b^	49.13 ± 6.40 ^ab^	50.17 ± 7.10 ^ac^	51.85 ± 6.38 ^c^
Shape indices					
Ellipticity	1.97 ± 0.28 ^a^	1.99 ± 0.33 ^a^	1.90 ± 0.28 ^bc^	1.96 ± 0.31 ^a^	1.86 ± 0.27 ^c^
Elongation	0.32 ± 0.06 ^a^	0.32 ± 0.07 ^a^	0.30 ± 0.06 ^b^	0.32 ± 0.07 ^a^	0.29 ± 0.06 ^b^
Rugosity	0.77 ± 0.06 ^a^	0.76 ± 0.07 ^ab^	0.75 ± 0.07 ^b^	0.78 ± 0.07 ^a^	0.78 ± 0.06 ^a^
Regularity	0.89 ± 0.16 ^a^	0.92 ± 0.17 ^ac^	0.98 ± 0.17 ^b^	0.95 ± 0.15 ^bc^	0.94 ± 0.15 ^bc^

^a,b,c^ Different letters in rows indicate the differences (*p* ≤ 0.05).

**Table 2 animals-12-01888-t002:** Morphometric dimensions of sperm stained by the eosin-gentian method in semen stored at 17 °C (mean ± SD).

Item	Semen Storage Time (Hour)				
	1	24	48	96	168
Number of analyzed cells	1200	1200	1200	1200	1200
Head					
Length (µm)	9.21 ± 0.78 ^a^	9.45 ± 0.78 ^b^	9.63 ± 0.80 ^b^	9.65 ± 0.77 ^b^	9.46 ± 0.72 ^b^
Width (µm)	4.85 ± 0.61 ^a^	4.99 ± 0.72 ^a^	5.20 ± 0.65 ^bc^	5.13 ± 0.69 ^c^	4.98 ± 0.76 ^a^
Area (µm^2^)	38.89 ± 4.17 ^a^	39.68 ± 4.53 ^ab^	40.21 ± 4.64 ^b^	40.20 ± 4.75 ^b^	39.56 ± 4.79 ^ab^
Perimeter (µm)	24.92 ± 1.38 ^a^	25.10 ± 1.47 ^ab^	25.53 ± 1.80 ^bc^	25.77 ± 1.77 ^c^	25.54 ± 2.33 ^bc^
Tail					
Length (µm)	44.93 ± 2.89 ^a^	44.66 ± 4.28 ^a^	45.29 ± 4.87 ^a^	45.32 ± 4.27 ^a^	45.70 ± 4.38 ^a^
Sperm total length (µm)	54.14 ± 3.09 ^a^	54.11 ± 4.28 ^a^	54.93 ± 4.95 ^a^	54.97 ± 4.28 ^a^	55.17 ± 4.38 ^a^
Shape indices					
Ellipticity	1.93 ± 0.29 ^a^	1.93 ± 0.32 ^a^	1.88 ± 0.26 ^a^	1.93 ± 0.30 ^a^	1.94 ± 0.31 ^a^
Elongation	0.31 ± 0.06 ^a^	0.31 ± 0.07 ^a^	0.30 ± 0.06 ^a^	0.31 ± 0.07 ^a^	0.31 ± 0.08 ^a^
Rugosity	0.81 ± 0.05 ^a^	0.78 ± 0.06 ^b^	0.78 ± 0.08 ^b^	0.75 ± 0.08 ^c^	0.77 ± 0.08 ^b^
Regularity	0.88 ± 0.13 ^a^	0.96 ± 0.17 ^b^	0.99 ± 0.16 ^b^	0.99 ± 0.15 ^b^	0.94 ± 0.14 ^ab^

^a,b,c^ Different letters in rows indicate the differences (*p* ≤ 0.05).

**Table 3 animals-12-01888-t003:** Morphometric dimensions of sperm stained by the SpermBlue method in semen stored at 17 °C (mean ± SD).

Item	Semen Storage Time (Hour)				
	1	24	48	96	168
Number of analyzed cells	1200	1200	1200	1200	1200
Head					
Length (µm)	9.11 ± 0.10 ^a^	9.10 ± 0.84 ^a^	9.31 ± 0.78 ^ab^	9.47 ± 0.90 ^b^	9.29 ± 0.78 ^ab^
Width (µm)	4.80 ± 0.72 ^ac^	4.76 ± 0.73 ^a^	5.02 ± 0.72 ^b^	4.91 ± 0.65 ^ab^	4.98 ± 0.59 ^bc^
Area (µm^2^)	37.72 ± 4.28 ^a^	35.67 ± 4.42 ^b^	38.18 ± 4.88 ^a^	38.52 ± 4.43 ^a^	37.89 ± 4.75 ^a^
Perimeter (µm)	24.84 ± 1.85 ^a^	24.24 ± 1.74 ^b^	25.25 ± 2.02 ^ac^	25.38 ± 1.75 ^c^	25.03 ± 1.65 ^ac^
Tail					
Length (µm)	45.74 ± 4.37 ^a^	44.55 ± 4.80 ^a^	45.30 ± 4.98 ^a^	44.38 ± 6.17 ^a^	45.40 ± 4.97 ^a^
Sperm total length (µm)	54.85 ± 4.65 ^a^	53.65 ± 4.93 ^a^	54.60 ± 5.14 ^a^	53.85 ± 6.43 ^a^	54.69 ± 5.18 ^a^
Shape indices					
Ellipticity	1.93 ± 0.30 ^a^	1.96 ± 0.35 ^a^	1.88 ± 0.27 ^a^	1.96 ± 0.30 ^a^	1.89 ± 0.28 ^a^
Elongation	0.31 ± 0.07 ^a^	0.31 ± 0.08 ^a^	0.30 ± 0.06 ^a^	0.32 ± 0.07 ^a^	0.30 ± 0.07 ^a^
Rugosity	0.77 ± 0.07 ^a^	0.76 ± 0.08 ^a^	0.76 ± 0.09 ^a^	0.75 ± 0.08 ^a^	0.76 ± 0.07 ^a^
Regularity	0.92 ± 0.20 ^a^	0.96 ± 0.17 ^a^	0.97 ± 0.17 ^a^	0.95 ± 0.15 ^a^	0.97 ± 0.15 ^a^

^a,b,c^ Different letters in rows indicate the differences (*p* ≤ 0.05).

## Data Availability

No new data were created or analyzed in this study. Data sharing is not applicable to this article.

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
