# Peer review of "Morphometry of Boar Spermatozoa in Semen Stored at 17 °C—The Influence of the Staining Technique"

_animals, 2022, doi:10.3390/ani12151888_

Round 1

Reviewer 1 Report

The manuscript describes a study on the morphometry of boar spermatozoa during storage of insemination doses comparing three staining techniques. The authors conclude that sperm morphometry does change during storage but differences are seen according to which staining technique is used.

Information in the Introduction regarding sperm cryopreservation is incorrect where it states that cryopreservation is not possible. There are practical methods available for cryopreserving boar sperm that are used for artificial insemination, e.g. when sending semen doses to other countries. However, the farrowing rate and litter size tend to be lower than when using non-frozen semen. Since the margins in the pig breeding industry are so small, pig breeders cannot afford to have fewer sows farrowing or smaller litter sizes born and therefore they choose to use non-frozen semen.

I have several problems with this study. Many studies over the years have shown a relationship between sperm morphology and fertility but differing opinions have arisen regarding the relationship between sperm morphometry and fertility.  Some authors consider that morphometry is linked to fertility; others suggest that more work is needed before any conclusions can be drawn. This begs the question – why are the authors interested in boar sperm morphometry? Do they have any evidence that morphometry changes during storage are linked to boar fertility? What was the fertility of the boars used as semen donors in this study?

Is it enough to make morphometry measurements on 15 sperm from an insemination dose when each dose contains 2.9 billion sperm? Why choose 15? This should be explained and referenced. Why choose these three stains? There are others more commonly used for morphology, such as carbol-fuchsin.

The staining protocol is not clear: why was the temperature of the stain “about 37°C”, especially when using semen stored at 17°C? Would it not be better to use the stain at the same temperature as the semen? Why “about” – do the authors not know what the temperature was? Could the difference in morphometry between 1 h and 24 h be due only to the starting temperature of the material, assuming that at 1 h the sperm samples were at ambient temperature as they had not yet been cooled to 17°C? If this assumption is true, what was the ambient temperature? Have the authors done a control to see if the starting temperature of the semen at 1 h affects morphometry, or conversely, if allowing the stored semen to stand at ambient temperature before staining makes a difference to the morphometry?

Is there another difference between the stains other than the fixing/not fixing step when making the smears? What about pH, osmolarity and general toxicity of the stains? These factors and their potential effect on sperm morphometry should be discussed.

I miss a proper conclusion: are the authors saying that they see differences in morphometry during storage but this is due to a staining artefact? Or are they saying that the changes would be seen anyway and have a biological significance? If so, what is the significance in the absence of any information about the fertility of the boars and the small number of sperm counted? If sperm morphometry is confounded by staining technique, is it reliable as an indicator of anything?

The morphometry characteristic that has previously been linked to fertility and reported in the literature is that smaller sperm heads are associated with higher fertility. This observation has been explained in relation to chromatin packaging, as mentioned by the authors in the Discussion. The more tightly the chromatin is packaged, the less opportunity there is for DNA strand breaks to occur. Chromatin packaging is seen to change during storage of boar sperm, particularly at or after day 3. How do the authors reconcile their morphometry results in the present study with these published findings? Maybe it would be more meaningful to assess morphometry together with chromatin packaging? This might help to resolve the current dilemma about what the comparison of the staining techniques actually means.

Author Response

Thank you for your valuable comments, which will certainly contribute to improving the quality of the manuscript. I hope that our changes to the text and responses to the comments will be appropriate and that the manuscript can be published in Animals. All the changes introduced to the text are highlighted with a red colour.

Response to comments from the Reviewer 1

Information in the Introduction regarding sperm cryopreservation is incorrect where it states that cryopreservation is not possible. There are practical methods available for cryopreserving boar sperm that are used for artificial insemination, e.g. when sending semen doses to other countries. However, the farrowing rate and litter size tend to be lower than when using non-frozen semen. Since the margins in the pig breeding industry are so small, pig breeders cannot afford to have fewer sows farrowing or smaller litter sizes born and therefore they choose to use non-frozen semen.

Response

Thank you for this valuable comment. Our statement regarding preservation was inaccurate. Of course there are methods of cryopreservation of boar semen, but due to the low sperm survival rate, these methods are not commonly used. We have corrected the information in the Introduction (Lines 50-52).

I have several problems with this study. Many studies over the years have shown a relationship between sperm morphology and fertility but differing opinions have arisen regarding the relationship between sperm morphometry and fertility. Some authors consider that morphometry is linked to fertility; others suggest that more work is needed before any conclusions can be drawn. This begs the question – why are the authors interested in boar sperm morphometry? Do they have any evidence that morphometry changes during storage are linked to boar fertility? What was the fertility of the boars used as semen donors in this study?

Response

Thank you very much for your comments.

The relationship between the morphometric dimensions of sperm and male fertility has been studied by many authors. Some believe that such a relationship exists, while others have not found evidence of it. Moreover, such studies are often conducted on fresh semen.  There are no studies evaluating sperm morphometry during storage of liquid boar semen. Due to the structure of the cell membrane of boar sperm, various changes can take places in individual morphological structures. Therefore we thought it was worth testing whether sperm morphometry changes during storage of boar semen. As it is not entirely clear whether sperm morphometry is associated with fertility, we attempted to assess the morphometric dimensions of boar sperm in relation to storage time. Fertility assessment would undoubtedly be an excellent addition to the research, and we will certainly consider this in the future. The boars chosen for the study were used for artificial insemination and the ejaculates were tested after each collection. The breeders who used them did not report any unsuccessful insemination procedures. 

Is it enough to make morphometry measurements on 15 sperm from an insemination dose when each dose contains 2.9 billion sperm? Why choose 15? This should be explained and referenced. Why choose these three stains? There are others more commonly used for morphology, such as carbol-fuchsin.

Response

Thank you very much for this suggestion. We made morphometric measurements in 15 sperm with normal morphology (three slides from each insemination dose, stained by different methods). In each sperm we made five measurements (head length, width, perimeter, and area and tail length; total sperm length was calculated as the sum of the head length and tail length). These tests require a great deal of work, and for this reason fewer cells are used in the case of assessment of morphometry. Of course the more cells are examined, the more accurate the assessment is. In our study we used 20 individuals from one breed, so we believed that morphometric measurements of 15 sperm per slide (45 sperm per insemination dose – 15 for each staining method) were sufficient to ensure accurate evaluation of sperm morphometry.

We chose three staining methods: eosin-nigrosin, eosin-gentian and SpermBlue. We chose the eosin-nigrosin method because it is also used for assessment of live and dead cells and does not require fixing in any buffers or alcohol. We have been using the eosin-gentian method for many years to evaluate the morphology and morphometry of sperm of various animal species, and we have observed that it thoroughly stains the head and tail, which is very important in morphometric measurements. The SpermBlue method is often used to evaluate sperm morphometry using the SCA system, so we wanted to compare this method with the others. There are other commonly used methods, especially in evaluation of sperm morphology, such as carbol-fuchsin. We refer to other methods in the Discussion (lines 245-266, 274-281).

The staining protocol is not clear: why was the temperature of the stain “about 37°C”, especially when using semen stored at 17°C? Would it not be better to use the stain at the same temperature as the semen? Why “about” – do the authors not know what the temperature was? Could the difference in morphometry between 1 h and 24 h be due only to the starting temperature of the material, assuming that at 1 h the sperm samples were at ambient temperature as they had not yet been cooled to 17°C? If this assumption is true, what was the ambient temperature? Have the authors done a control to see if the starting temperature of the semen at 1 h affects morphometry, or conversely, if allowing the stored semen to stand at ambient temperature before staining makes a difference to the morphometry?

Response

Thank you for these valuable comments. We have deleted the word ‘about’, which was used mistakenly. The microscope slides on which the smears were prepared were heated to 37°C, because it is close to the animal’s body temperature. It is recommended to use heated slides for these methods. The temperature in the laboratory was 22°C. We wanted the temperature at which the insemination doses were stored to be the same at every hour (17°C). Immediately before preparing the smears, the insemination doses were taken out of the refrigerator, where the temperature was 17°C.

Is there another difference between the stains other than the fixing/not fixing step when making the smears? What about pH, osmolarity and general toxicity of the stains? These factors and their potential effect on sperm morphometry should be discussed.

Response

Various factors can influence sperm morphometry, such as smear preparation time, reagent composition, fixing, the level of experience of the person performing the measurement, the pH of the stains, etc. We have discussed these factors in the Discussion (lines 267-272)

I miss a proper conclusion: are the authors saying that they see differences in morphometry during storage but this is due to a staining artefact? Or are they saying that the changes would be seen anyway and have a biological significance? If so, what is the significance in the absence of any information about the fertility of the boars and the small number of sperm counted? If sperm morphometry is confounded by staining technique, is it reliable as an indicator of anything?

Response

Thank you for these comments.

We have revised the conclusion to say that changes take place in sperm morphometry during storage of liquid boar semen. These differences are visible in each of the staining techniques.

The morphometry characteristic that has previously been linked to fertility and reported in the literature is that smaller sperm heads are associated with higher fertility. This observation has been explained in relation to chromatin packaging, as mentioned by the authors in the Discussion. The more tightly the chromatin is packaged, the less opportunity there is for DNA strand breaks to occur. Chromatin packaging is seen to change during storage of boar sperm, particularly at or after day 3. How do the authors reconcile their morphometry results in the present study with these published findings? Maybe it would be more meaningful to assess morphometry together with chromatin packaging? This might help to resolve the current dilemma about what the comparison of the staining techniques actually means.

Response

Thank you very much for these valuable suggestions. I think they will be useful to us in our future research. In this study we wanted to test three staining techniques in the assessment of sperm morphometry during semen storage. Other studies have tested the effect of various factors on sperm quality during semen storage, but not sperm morphometry. We know from other studies that the morphometric dimensions of sperm may indicate various abnormalities, e.g. in the head. 

Reviewer 2 Report

the manuscript is important and relevant, the role of morphology and morphometry in evaluating sperm fertilizing ability is well known for about a century (see Williams and Savage, Cornell Vet., 17 (1927), pp. 374-385). I have the feeling however, that Authors should have dug deeper in the relevant literature of other species. Morphometry of sperm cells have a growing presence in the field of evolutionary biology (see https://spermtree.org/, for ex. or the book of B Jamieson, Fish Evolution and Systematics: Evidence from Spermatozoa, Cambridge University Press, 1992).

Authors fail to explain the dynamic changes of sperm dimensions over time their discussion is rather superficial. Indeed, necrotic plasma membrane and acrosomal membrane changes have a direct effect on the 2D measurements of sperm heads, see Revay et al., Reprod Fertil Dev. 2004;16(7):681-7. 

Finally, Authors fail to refer to some similar studies on the effects of different staining techniques on sperm morphology, like Czubaszek et al., PLoS
ONE 14(3): e0214243. 2019, or even Boersma et al., Reprod Dom Anim 36, 222-229, 2001.

Minor comment: a typo in the table footnotes: Different letters in rows indicato differences

In conclusion, the study needs a thorough major revision.

Author Response

Thank you for your valuable comments, which will certainly contribute to improving the quality of the manuscript. I hope that our changes to the text and responses to the comments will be appropriate and that the manuscript can be published in Animals. Thank you for your valuable comments, which will certainly contribute to improving the quality of the manuscript. I hope that our changes to the text and responses to the comments will be appropriate and that the manuscript can be published in Animals. All the changes introduced to the text are highlighted with a red colour.

Response to comments from the Reviewer 2

The manuscript is important and relevant, the role of morphology and morphometry in evaluating sperm fertilizing ability is well known for about a century (see Williams and Savage, Cornell Vet., 17 (1927), pp. 374-385). I have the feeling however, that Authors should have dug deeper in the relevant literature of other species. Morphometry of sperm cells have a growing presence in the field of evolutionary biology (see https://spermtree.org/, for ex. or the book of B Jamieson, Fish Evolution and Systematics: Evidence from Spermatozoa, ‎ Cambridge University Press, 1992).

Authors fail to explain the dynamic changes of sperm dimensions over time their discussion is rather superficial. Indeed, necrotic plasma membrane and acrosomal membrane changes have a direct effect on the 2D measurements of sperm heads, see Revay et al., Reprod Fertil Dev. 2004;16(7):681-7. 

Finally, Authors fail to refer to some similar studies on the effects of different staining techniques on sperm morphology, like Czubaszek et al., PLoSONE 14(3): e0214243. 2019, or even Boersma et al., Reprod Dom Anim 36, 222-229, 2001.

Response

Thank you very much for your comments, which have inspired us to improve our work.

In the Discussion we have added information on sperm morphometry, changes in morphometric dimensions with semen storage time, and the effect of various staining techniques (Lines 245-272, 300-301, 324-338).

We have added the following literature references:

  1. Henkel, R.; Schreiber, G.; Sturmhoefel, A.; Hipler, U.C.; Zermann, D.H.; Menkveld, R. Comparison of three staining methods for the morphological evaluation of human spermatozoa. Fertil Steril. 2008, 89, 449–455, https://doi.org/10.1016/j.fertnstert.2007.03.027
  2. Chacon, J. Assessment of Sperm Morphology in Zebu Bulls, under Field Conditionsin the Tropics. Dom. Anim. 2001, 36, 91-99, https://doi.org/10.1046/j.1439-0531.2001.00253.x
  3. Boersma, A.; Raûhofer, R.; Stolla, R. Influence of Sample Preparation, Staining Procedure and Analysis Conditions on Bull Sperm Head Morphometry using the Morphology Analyser Integrated Visual Optical System. Dom. Anim. 2001, 36, 222-229, DOI: 10.1046/j.1439-0531.2001.00291.x
  4. Hidalgo, M.; Rodriguez, I.; Dorado, J.; Sanz, J.; Soler, C. Effect of sample size and staining methods on stallion sperm morphometry by the Sperm Class Analyzer. Med. 2012, 50, 24–32, doi:10.17221/5593‐vetmed
  5. Czubaszek, M.; Andraszek, K.; Banaszewska, D.; Walczak-Jędrzejowska, R. The effect of the staining technique on morphological and morphometric parameters of boar sperm. PLoS ONE 2019, 14 (3), 1–17, https://doi.org/10.1371/journal.pone.
  6. Łukaszewicz, E; A. Jerysz, A.; Partyka, A.; Siudzinska, A. Efficacy of evaluation of rooster sperm morphology using different staining methods. Vet. Sci. 2008, 85, 583–588, doi:10.1016/j.rvsc.2008.03.010
  7. Gacem, S.; Catalán, J.; Yánez‐Ortiz, I.; Soler, C.; Miró, J. New Sperm Morphology Analysis in Equids: Trumorph® Vs Eosin‐Nigrosin Stain. Sci. 2021, 8, 79, https://doi.org/10.3390/vetsci8050079
  8. Kahrl, A.F., Snook, R.R.; Fitzpatrick, J.L. Fertilization mode drives sperm length evolution across the animal tree of life. Ecol. Evol. 2021, 5, 1153–1164, https://doi.org/10.1038/s41559-021-01488-y
  9. Fitzpatrick, J.L.; Bridge, C.D.; Snook, R.R. Repeated evidence that the accelerated evolution of sperm is associated with their fertilization function. R. Soc. B 2020, 287, 20201286, http://dx.doi.org/10.1098/rspb.2020.1286
  10. Maxwell, W.M.C.; Johnson, L.A. Physiology of spermatozoa at high dilution rates: the influence of seminal plasma Theriogenology 1999, 52, 1353-1362, DOI: 10.1016/s0093-691x(99)00222-8
  11. Boe-Hansen, G.B.; Annette, K.; Ersbøll A.K.; Greve, T.; Christensen, P. Increasing storage time of extended boar semen reduces sperm DNA integrity. Theriogenology 2005, 63, 2006–2019 doi:10.1016/j.theriogenology.2004.09.006
  12. Révay, T.; Nagy, S.; Kovács, A.; Edvi, M.E.; Hidas, A.; Rens, W.; Gustavsson, I. Head area measurements of dead, live, X- and Y-bearing bovine spermatozoa. Fertil.Dev. 2004, 16, 681–687, doi: 10.1071/rd04013
  13. Saini, J.; Dhande, P.L.; Gaikwad, S.A.; Shankhapal, V.D.; Hmangaihzuali, E.V.L.; Walters, A. Comparative study on sperm morphology and morphometry of Holstein friesian and murrah buffalo bull. Buffalo Bulletin 2018, 37 (4), 559-567.

Minor comment: a typo in the table footnotes: Different letters in rows indicato differences

Response

We have corrected ‘indicato’ to ‘indicate’.

Reviewer 3 Report

This study evaluated boar sperm morphometry during storage of liquid boar semen using three staining methods and compared the results from these methods. The introduction is concise and well written, and the results are well displayed. However, there are some points that need to be addressed. These points concern the comparison of the three staining method, as there is no control (unstained sperm) group, which makes it impossible to draw conclusions on the suitability of these methods, and the time factor in this study is not discussed enough. The novelty in this paper is the time factor, but unfortunately, because there is no control group, it is hard to draw conclusions and discuss the results properly. In my opinion, it is necessary to add a control/unstained group for more scientific soundness.

General comments:

1.     Could you give an explanation on why you specifically chose eosin-nigrosin, eosin-gentian, and SpermBlue stainings only?

2.     What is the biological reasoning for choosing 1, 24, 48, 96, and 168 hours?

3.     Lines 92-93. Which methods were used to evaluate those parameters? Where was the semen collected (in graduated tubes?)

4.     Line 93. Were the samples washed before?

5.     Lines 134, 143, 147, 151, 166, 170, 176, 216. P value should be in lower case italics

6.     Line 120 and 150. Is it the number of cells per animal or in general (from 80 samples)?

7.     There is a lack of discussion about eosin-nigrosin staining method and why such differences were found between this method and the other methods. Is there a chemical that might influence the shape of the head? Is this chemical present in eosin or nigrosin? (probably in nigrosin since one of the other method-EG- has eosin as well), or is it a problem in the other methods?

8.     Line 265. From these results, it seems that eosin-nigrosin staining is not suitable for a proper morphometry evaluation, or is it that the other methods induce a swelling in the head part and therefore are not suitable for a morphometry evaluation? There is no discussion about which methods seem to be more trustworthy. The best method should be the one that interferes the least with the sperm structure and size, but unfortunately, because there is no control group, it is impossible to draw conclusions from these results only. According to previous studies that used a control group, which one of these methods seems more ideal?

9.     It’s the same for the second paragraph in the discussion: the time is mentioned only once at the end of the paragraph but the discussion is mainly about the importance of the head shape. What about the effect of time? And how does it influence sperm morphometrics? As stated in the abstract, the aim of the study was to assess morphometry of sperm during storage of liquid boar semen at 17 degrees, and the morphometry parameters were assessed at different times, but there is no discussion about the semen storage time and its influence on morphometrics, the time is only mentioned twice in the whole discussion (Line 237 and line 281), otherwise what is the utility of measuring those parameters at different times. Which time seems to be ideal for an insemination? 

Author Response

Thank you for your valuable comments, which will certainly contribute to improving the quality of the manuscript. I hope that our changes to the text and responses to the comments will be appropriate and that the manuscript can be published in Animals. Thank you for your valuable comments, which will certainly contribute to improving the quality of the manuscript. I hope that our changes to the text and responses to the comments will be appropriate and that the manuscript can be published in Animals. All the changes introduced to the text are highlighted with a red colour.

Response to comments from the Reviewer 3

This study evaluated boar sperm morphometry during storage of liquid boar semen using three staining methods and compared the results from these methods. The introduction is concise and well written, and the results are well displayed. However, there are some points that need to be addressed. These points concern the comparison of the three staining method, as there is no control (unstained sperm) group, which makes it impossible to draw conclusions on the suitability of these methods, and the time factor in this study is not discussed enough. The novelty in this paper is the time factor, but unfortunately, because there is no control group, it is hard to draw conclusions and discuss the results properly. In my opinion, it is necessary to add a control/unstained group for more scientific soundness.

General comments:

  1. Could you give an explanation on why you specifically chose eosin-nigrosin, eosin-gentian, and Sperm Blue stainings only?

Response

Thank you very much for your comments

We chose three staining methods: eosin-nigrosin, eosin-gentian and SpermBlue. We chose the eosin-nigrosin method because it is also used for assessment of live and dead cells and does not require fixing in any buffers or alcohol. We have been using the eosin-gentian method for many years to evaluate the morphology and morphometry of sperm of various animal species, and we have observed that it thoroughly stains the head and tail, which is very important in morphometric measurements. The SpermBlue method is often used to evaluate sperm morphometry using the SCA system, so we wanted to compare this method with the others. There are other commonly used methods, especially in evaluation of sperm morphology, such as carbol-fuchsin.

We refer to other methods in the Discussion (lines 273-285 and 245-266).

  1. What is the biological reasoning for choosing 1, 24, 48, 96, and 168 hours?

Response

Based on our own observations and literature data, we concluded that functional and structural changes can take place in sperm during storage of liquid boar semen. Boar sperm are known to be highly sensitive to cooling and storage conditions. In artificial insemination practice, most insemination doses are used within 48 hours after collection, but some may be used up to five days later. The ejaculates collected for the present study were diluted with Biosolwens Plus, a long-term extender which maintains sperm motility even up to eight days. Therefore, knowing that various changes can take place in the sperm head and midpiece during storage of liquid boar semen (as shown in our previous research), we assessed the sperm after1, 24, 48, 96 and 168 hours.

  1. Lines 92-93. Which methods were used to evaluate those parameters? Where was the semen collected (in graduated tubes?)

Response

We have added Material and methods (Lines 92-99)

  1. Line 93. Were the samples washed before?

The samples were diluted with Biosolwens Plus.

  1. Lines 134, 143, 147, 151, 166, 170, 176, 216. P value should be in lower case italics

Response

Corrected P

  1. Line 120 and 150. Is it the number of cells per animal or in general (from 80 samples)?

Response

Line 120 now line 124 It is number of cells from 80 ejaculates (four ejaculates per boar x 20 boars x 3 staining methods x 5 semen storage time x 15 spermatozoa)

Line 150 now line 154 It is number of cells from 80 ejaculates (four ejaculates per boar x 20 boars x 1 staining methods x 1 semen storage time x 15 spermatozoa)

  1. There is a lack of discussion about eosin-nigrosin staining method and why such differences were found between this method and the other methods. Is there a chemical that might influence the shape of the head? Is this chemical present in eosin or nigrosin? (probably in nigrosin since one of the other method-EG- has eosin as well), or is it a problem in the other methods?

Response

Thank you for this valuable comment.  We agree that we included too little information on the eosin-nigrosin method in the Discussion. This is a simple method that has been widely used in assessment of human and animal sperm morphology. We have provided more information on this method and the other methods used in the study (Lines 274-281)

  1. Line 265. From these results, it seems that eosin-nigrosin staining is not suitable for a proper morphometry evaluation, or is it that the other methods induce a swelling in the head part and therefore are not suitable for a morphometry evaluation? There is no discussion about which methods seem to be more trustworthy. The best method should be the one that interferes the least with the sperm structure and size, but unfortunately, because there is no control group, it is impossible to draw conclusions from these results only. According to previous studies that used a control group, which one of these methods seems more ideal?

Response

Thank you for this. We have added information to the Discussion about the staining methods and their effect on sperm morphology. We did not refer to a control group (unstained smears) because morphometric measurements must be exact. Individual parts of the sperm cell must be very clearly visible for the evaluation to be reliable. In unstained smears the measurements are less precise. (Lines 245-266) 

  1. It’s the same for the second paragraph in the discussion: the time is mentioned only once at the end of the paragraph but the discussion is mainly about the importance of the head shape. What about the effect of time? And how does it influence sperm morphometrics? As stated in the abstract, the aim of the study was to assess morphometry of sperm during storage of liquid boar semen at 17 degrees, and the morphometry parameters were assessed at different times, but there is no discussion about the semen storage time and its influence on morphometrics, the time is only mentioned twice in the whole discussion (Line 237 and line 281), otherwise what is the utility of measuring those parameters at different times. Which time seems to be ideal for an insemination? 

Response

Thank you for this valuable comment.

In the Discussion section, we've added more information on the effect of boar semen storage time (lines 330-338). This is an extremely important aspect of liquid preserved boar semen. Some believe that there is a link between sperm morphometry and male fertility. Moreover, such studies are often conducted on fresh semen.  There are no studies evaluating sperm morphometry during storage of liquid boar semen. Due to the structure of the cell membrane of boar sperm, various changes can take places in individual morphological structures. Therefore we thought it was worth testing whether sperm morphometry changes during storage of boar semen. As it is not entirely clear whether sperm morphometry is associated with fertility, we attempted to assess the morphometric dimensions of boar sperm in relation to storage time. In the future, we plan to conduct such research, but also take into account fertility.

Round 2

Reviewer 1 Report

Unfortunately the authors have not addressed all of my original comments, or have explained them in their response but not added the information to the manuscript. The reason for providing my comments is so that the manuscript can be explained better for the benefit of the readers. So, please, add your explanation TO THE MANUSCRIPT.

Add to the Introduction why you consider that it is necessary to study morphometry in stored boar semen samples. The paragraph that you have added to the Discussion (lines 330-338) would have made an ideal Introduction as to why you think this topic should be studied. I suggest that you move these lines to the introduction and show in the Discussion that this is indeed the case (if you can……).

Introduce the stains that you have chosen to use in the introduction. Why these particular stains and not others that are available?

In the methods section, add information about the fertility of the boars. It is not enough to state that they were routinely used at the AI station. Many AI stations (mistakenly) do not know the fertility of individual boars since the semen doses contain a mixture of semen from several boars. However, in the case of your study, you (and the reader) need to know the fertility of the boars, especially since you claim that the differences in fertility can be detected by morphometry. This is key to your study so please add this information TO THE MANUSCRIPT.

So the samples had already been cooled to 17 °C for 1 h before the first assessment. Make this clear IN THE MANUSCRIPT. At present it says that the evaluation was done 1h after semen collection. In your response you say that semen had already been cooled for 1h, so which is it? Please make this clear.

I still miss a proper conclusion -  see my original comment

“ I miss a proper conclusion: are the authors saying that they see differences in morphometry during storage but this is due to a staining artefact? Or are they saying that the changes would be seen anyway and have a biological significance? If so, what is the significance in the absence of any information about the fertility of the boars and the small number of sperm counted? If sperm morphometry is confounded by staining technique, is it reliable as an indicator of anything?”

Author Response

Response to comments from the Reviewer1

Thank you very much for re-reviewing my article and your time. I hope that our changes to the text and responses to the comments will be appropriate and that the manuscript can be published in Animals. All the changes introduced to the text are highlighted with the red colour.

Unfortunately the authors have not addressed all of my original comments, or have explained them in their response but not added the information to the manuscript. The reason for providing my comments is so that the manuscript can be explained better for the benefit of the readers. So, please, add your explanation TO THE MANUSCRIPT.

Response

I apologise for being unable to address all of the comments and include them in the manuscipt. I will try to make my answers and additions as appropriate to the comments as possible and improve the quality of the manuscript. Thank you very much for your valuable comments.

Add to the Introduction why you consider that it is necessary to study morphometry in stored boar semen samples. The paragraph that you have added to the Discussion (lines 330-338) would have made an ideal Introduction as to why you think this topic should be studied. I suggest that you move these lines to the introduction and show in the Discussion that this is indeed the case (if you can……).

Response

Thank you very much for these comments. The sentences „Most diluted insemination doses are used for artificial insemination on the day they are collected, but some are used several days later [7]. Dilution of boar semen is believed to reduce proteins and natural antioxidants together with other components of seminal plasma which are essential to the proper functioning and integrity of sperm membranes [8]. During storage of boar semen, functional and structural changes take place in sperm, which can be affected by the dilution conditions and storage time [9]. For this reason it is important to monitor boar semen during storage to the extent possible” transferred to Introduction (Lines 56-63).

The discussion chapter has been supplemented (Lines 309-315)

Introduce the stains that you have chosen to use in the introduction. Why these particular stains and not others that are available?

Response

The Introduction was supplemented with the characteristics of the staining methods used for the research (Lines 78-88)

In the methods section, add information about the fertility of the boars. It is not enough to state that they were routinely used at the AI station. Many AI stations (mistakenly) do not know the fertility of individual boars since the semen doses contain a mixture of semen from several boars. However, in the case of your study, you (and the reader) need to know the fertility of the boars, especially since you claim that the differences in fertility can be detected by morphometry. This is key to your study so please add this information TO THE MANUSCRIPT.

Response

Information on the boars fertility (Line 104) has been supplemented in the Materials and Methods.

So the samples had already been cooled to 17 °C for 1 h before the first assessment. Make this clear IN THE MANUSCRIPT. At present it says that the evaluation was done 1h after semen collection. In your response you say that semen had already been cooled for 1h, so which is it? Please make this clear.

Response

Thank you very much for this comment. I apologise for the unfortunate wording in the previous answer. I hope you will find my explanation in the Material and methods relevant (Lines 122-123).

I still miss a proper conclusion -  see my original comment

“ I miss a proper conclusion: are the authors saying that they see differences in morphometry during storage but this is due to a staining artefact? Or are they saying that the changes would be seen anyway and have a biological significance? If so, what is the significance in the absence of any information about the fertility of the boars and the small number of sperm counted? If sperm morphometry is confounded by staining technique, is it reliable as an indicator of anything?”

Response

Thank you very much for this comment. Changes have been made to Conclusion (Lines 356-362). I hope these changes are appropriate.

Reviewer 2 Report

Authors followed my suggestions and extended the cited relevant literature and incorporated these into the discussion, therefore in my opinion the manuscript is acceptable with minor changes without further review. Minor changes are mainly typo-s in the new paragraphs, please check these. For ex.:

lines 250-251: consumingand

line 260: AgNO3staining

Author Response

Response to comments from the Reviewer 2

Thank you very much for reviewing my article.

Comments and Suggestions for Authors

Authors followed my suggestions and extended the cited relevant literature and incorporated these into the discussion, therefore in my opinion the manuscript is acceptable with minor changes without further review. Minor changes are mainly typo-s in the new paragraphs, please check these. For ex.:

lines 250-251: consumingand

Response

corrected

line 260: AgNO3staining

Response

corrected

Reviewer 3 Report

The comments were all addressed well and the revised version seems to be suitable for publication

Author Response

Response to comments from the Reviewer 3

 Thank you very much for reviewing my article.